# "One Event, One City": Promoting the Loyalty of Marathon Runners to a Host City by Improving Event Service Quality

**Xiaoying Chen** [1], **Brian H. Yim** [2], **Ziqing Tuo** [3], **Liangjun Zhou** [3,*], **Ting Liu** [4,*] and **James J. Zhang** [5]

1   Sport Media School, Guangzhou Sport University, Guangzhou 510500, China; 11253@gzsport.edu.cn
2   Department of Foundations, Leadership and Administration, Kent State University, Kent, OH 44240, USA; hyim@kent.edu
3   School of Leisure Sport and Management, Guangzhou Sport University, Guangzhou 510500, China; annatuo1002@163.com
4   Department of Political Theory, Guangdong Vocational Institute of Sports, Guangzhou 510663, China
5   Department of Kinesiology, University of Georgia, Athens, GA 30602, USA; jamesz48@uga.edu
*   Correspondence: 11212@gzsport.edu.cn (L.Z.); liuting901228@163.com (T.L.);
    Tel.: +86-13380088446 (L.Z.); +86-15920367081 (T.L.)

**Abstract:** China has entered into a new developmental phase where the government can promote national fitness, the sports industry, and city tourism, simultaneously. Rapidly becoming the largest single sporting events in China, marathon events help facilitate this integration. However, event organizers tend to focus on winning bids for events instead of improving event services and increasing the loyalty of runners to the host city, which could benefit the city for sustainable development. In this study, the antecedents affecting marathon runner loyalty to the city hosting the annual hallmark event (i.e., destination loyalty) were investigated by studying a sample of 392 repeat runners during a recent HengQin International Marathon (HQ-M). Conducting structural equation modeling (SEM) analyses, the proposed model, integrating event service quality (ESQ), destination image (DI), satisfaction (SAT), and destination loyalty (DLOY), was confirmed. Nine hypothesized paths were identified among these four constructs, including six direct paths and three indirect paths. Meanwhile, mediation effects and serial mediation effects of DI and SAT between ESQ and DLOY were found. Good ESQ, positive DI, and high SAT were found both separately and collectively to enhance DLOY and have important implications for the strategic marketing of sporting events and sustaining city brand image.

**Keywords:** event service quality; destination image; satisfaction; destination loyalty; host city; repeat runners; integrated model; HengQin; marathon

## 1. Introduction

After the 2008 Beijing Olympic Games, China entered a new stage where the government began promoting national fitness [1], the sports industry [2], and city tourism [3], simultaneously. Since 2014, the Chinese government has successively enacted more than 18 policies to include the sports industry as a part of the efforts of transforming and upgrading the national economy. In 2016, the State Council of China released an outline for "Healthy China 2030," an initiative to promote health among all citizens [4]. Meanwhile, since 2018, the Chinese government has proposed strategies for promoting three major regions: Beijing-Tianjin-Hebei Region, Yangtze River Economic Belt, and Guangdong-Hong Kong-Macao Greater Bay Area. The core idea is to stimulate urban agglomeration in order to drive overall development in each region. This goal has pushed China's urban marketing strategy toward accelerated and fierce competition.

In the context of these social initiatives, China has seen an explosive increase in marathon events in recent years. These events promote urban marketing, facilitate growth in the sports industry, and encourage healthy activity. According to data from the China

Athletics Association (CAA), the number of marathon events in China increased from 51 events in 2014 to 1828 events in 2019, during which time 89% of the municipal cities in China hosted at least one event [5]. Although marathons have become the largest single sporting events in China, a problematic trend has developed: event organizers tend to focus more on winning bids to host an event instead of improving event service and, in turn, increasing the loyalty of runners to the event and the host city.

Scholars have explored event loyalty among participants [6,7]. However, from the perspective of city marketing, loyalty to the host city is more important than event loyalty. Investigating host city loyalty among marathon participants can help cities implement the concept of "hosting events to promote urban development," also known as "one event, one city" [8]. Maintaining consistency with previous studies, the term "destination loyalty" (DLOY) was used to refer to host city loyalty among marathon runners. DLOY refers to the intention to revisit a destination and the willingness to recommend it to others. It represents an emotional bond between tourists and a host city during a sporting event [9]. Because it can provide useful and reliable information about a destination, favorable word-of-mouth (WOM) recommendations constitute a form of marketing that attracts potential tourists [10].

Due to the significance of DLOY, many scholars have investigated its antecedents. Previous findings from tourism studies indicate that service quality positively influences DLOY directly and indirectly. According to Hallak et al. [11], perceived quality predicted DLOY. Other scholars have found that destination image (DI) [12] and satisfaction (SAT) [13] mediated the relationship between event service quality (ESQ) and DLOY. In the field of sport tourism, Jeong et al. [14] proposed that ESQ leads to behavioral intention and argued that destination marketers should strengthen ESQ in order to encourage sport tourists to revisit the host city. Likewise, if marathon runners enjoy high ESQ, they are more likely to come back to the host city and spread positive WOM to associates. Kaplanidou et al. [15] investigated marathon participants via an online survey to explore the association between DI and DLOY and found that DI predicted behavioral intention and WOM of active sport tourists. Moreover, SAT plays a pivotal role in the improvement of DLOY [16–19].

However, previous studies have three primary limitations. First, scholars have paid more attention to marathon events held in first- and second-tier cities, such as the Flora London Marathon [20] and Shanghai International Marathon [7,21]. However, the main force of rapid growth in China's marathon events continues to be in third- and fourth-tier cities. Therefore, the HengQin Marathon (HQ-M) held in Zhuhai was focused on.

Second, when addressing DLOY, scholars have not distinguished between new runners and repeat runners. It is proposed that a sample of repeat runners would increase the validity of these findings because repeat runners are more likely to have higher loyalty to events than new runners, who are more likely to abandon running due to various factors [22]. Meanwhile, based on previous findings, retaining regular customers is more cost-effective than soliciting new customers [23]. In the case of marathon events, repeat runners are retained customers. Therefore, the antecedent factors and the mechanism behind the formation of host city loyalty among repeat runners were explored.

Finally, although many scholars have explored the relationship between ESQ and DLOY, few have explored the pathway from ESQ to DLOY. Previous findings show that DI and SAT might each play an important role in this relationship, but no previous findings confirm that they might interact. To fill this gap, an integrated model including ESQ, DI, SAT, and DLOY was constructed.

Findings in this study make academic and practical contributions. DLOY was examined via ESQ, DI, and SAT, simultaneously, to expand the research areas of sport and city marketing and to identify a mediating mechanism between ESQ and DLOY. These findings should also help local authorities and event organizers facilitate the formation of DLOY by highlighting significant predictors of DLOY and pointing out strategies for destination brand sustainability.

## 2. Literature Review

### 2.1. Event Service Quality (ESQ)

ESQ is a measure of the overall judgment of and attitude toward the value or experience of an event attribute [14]. This topic is crucial in sport tourism research and has drawn attention from numerous scholars [24,25]. ESQ is an important precondition of the psychological and behavioral intentions of tourists [26]. Researchers have identified the dimensions of ESQ in numerous theoretical and measurement models. To measure service quality in a variety of sectors, scholars use the SERVQUAL model developed by Parasuraman et al. [27]. However, SERVQUAL is not suitable for outdoor settings [28]. Therefore, Shonk and Chelladurai [29] further modified the service quality of sport event tourism to include access, accommodation, venue, and contest quality. In the context of a small-scale running event, Theodorakis et al. [26] proposed a three-dimensional framework comprising physical environment quality, interaction quality, and outcome quality. In the context of sporting events, information quality (INF), facility quality (FAC), outcome quality, interaction quality (INT), and community attributes are five dimensions of ESQ [25]. To measure ESQ in the current study, the items of Huang et al. [25] and Theodorakis et al. [26] were used. Nine items were used to represent three subscales of ESQ: INF, INT, and FAC. INF is the feasibility of obtaining up-to-date information about services [30]. In this study, the measure of INF is about whether the information was easy to obtain online or on a mobile device and the convenience of registration. INT refers to the interaction between staff and clients [31]. Interactions between staff and runners can happen before, during, and after a running event and depend on the attitude, behavior, and expertise of employees [32]. In this case, INT includes the friendliness, knowledge, and response of staff and volunteers. According to Rust and Oliver [33], FAC is an important aspect of the service environment. However, facility type depends on the event type. As indicated by Huang et al. [25], in the case of a marathon, parking lots, shower rooms, transportation, cleanliness, bathrooms, medical facilities, and layout are potential measures of FAC. Therefore, the convenience of transportation, the number of aid stations, and facility layout were considered in this study.

### 2.2. Destination Image (DI)

DI is an essential concept in a wide range of studies [34]. The word "image" represents the sum of beliefs, impressions, ideas, and perceptions related to an object, behavior, or event [35]. In the context of tourism, DI is a combination of tourists' beliefs, ideas, and impressions related to a place they might visit for the first time or again. Scholars have conceptualized DI in various ways. Echtner and Ritchie [36] described DI using functional, attributional, holistic, and psychological characteristics. Baloglu and McCleary [37] developed a three-dimensional model, including cognitive, affective, and overall image. Using previous findings, Byon and Zhang [38] developed an original scale to measure DI, including items related to infrastructure, attraction, value for money, enjoyment, and behavioral intention. Distinguishing between stimulus factors and personal factors, Chen and Phou [39] reviewed relevant studies and found that scholars can measure DI based on the natural environment, atmosphere, and entertainment. Prayag and Ryan [40] used the exoticness of place to measure DI in the case of the island of Mauritius. Accordingly, the natural environment, atmosphere, entertainment, and exoticness of place DI dimensions were used in this study. Findings in the tourism and marketing literature suggest that enhancement of service quality positively influences DI [41–44]. For instance, MacCartney [45] tested the effect of ESQ on DI for a recurring mega event and found that ESQ played a significant role in engendering DI. In this way, sporting events are useful in improving awareness and DI of a region [44]. Therefore, the following hypothesis for the impact of ESQ on DI was proposed:

**Hypothesis 1 (H1).** *ESQ will positively influence DI.*

### 2.3. Satisfaction (SAT)

According to Sahin et al. [46], SAT is the emotional response to and positive emotional association with the outcome of an experience. Based on findings from marketing studies [19], SAT is the judgment reached after an evaluation and helps distinguish between expectation and authentic experience [47]. When clients perceive that service exceeds their expectations, they feel satisfied with the outcome [48]. In the context of tourism, SAT is the genuine feeling of tourists after interacting with the destination [49] and their overall rating of the entire travel process [50]. In terms of measurement, scholars have conceptualized overall SAT and its various attributes in the context of sport tourism. Overall, SAT refers to an evaluation of product purchasing and service delivery [51]. Attributes of SAT reflect the subjective judgments of tourists after experiencing or observing those attributes [52]. Other scholars have measured SAT by monitoring multiple items [53]. In the current study, three items proposed by Albaity and Melhem [54] were used to measure SAT: expectation–satisfaction, comparison with other places, and whether a place is worth visiting. Findings from tourism studies show that ESQ positively influences SAT [49,55]. When hosting an international sporting event, organizers should prioritize ESQ to satisfy sponsors and local residents [56]. In the view of Pantouvakis and Lymperopoulos [57], SAT depends on perceived ESQ. Lee and Su [58] also found that ESQ influenced SAT, suggesting that SAT is an indicator of ESQ [59]. Shonk and Chelladurai [29] found that sport ESQ directly contributed to SAT. Tourists are most likely to feel satisfied if they experience high-value products and services at a sporting event [23]. In addition, Greenwell and Fink [60] found that SAT primarily depended on ESQ service quality. Based on previous findings, the following hypothesis was proposed:

**Hypothesis 2 (H2).** *ESQ will positively influence SAT.*

Tourism findings also show that DI contributes to SAT [61,62]. Positive experiences with travel contribute to positive DI, and positive DI can lead to a more favorable evaluation of the destination. In other words, tourists are likely to feel higher SAT if they perceive a positive DI. For instance, in the context of tourism, Chi and Qu [10] set up a DI-SAT-DLOY model and found that DI related to SAT. Koo [63] found that DI positively influences SAT. Furthermore, Puh [59] investigated 705 tourists in Dubrovnik, Croatia, and found that DI influenced SAT. Based on these findings, the following hypothesis was proposed:

**Hypothesis 3 (H3).** *DI will positively influence SAT.*

### 2.4. Destination Loyalty (DLOY)

Loyalty is a useful tool for measuring the strength of consumer preference for a brand [64]. Customers are prepared to continue patronizing a brand and use the same products and services; they are also willing to recommend products and services to friends and relatives [65]. Reichheld and Sasser [66] found that retaining only 5% of clients could boost profits by nearly 100%. In tourism studies, DLOY is a key factor in gaining a competitive advantage and an effective strategy in tourism marketing [53]. However, DLOY is difficult to achieve [67].

When measuring DLOY, many scholars focus on two dimensions: intention to revisit the same destination (i.e., behavioral loyalty) and positive WOM [40,68,69]. Moreover, ESQ is an antecedent of DLOY. Jeong et al. [14] found that ESQ played a pivotal role in improving DLOY for a sporting event. Investigating marathon runners with high and low involvement, Alexandris et al. [32] found an association between ESQ and DLOY. According to Tzetzis, Alexandris, and Kapsampeli [70], ESQ helped increase DLOY for an outdoor sporting event. Based on these findings, the following hypothesis was proposed:

**Hypothesis 4 (H4).** *ESQ will positively influence DLOY.*

Most scholars agree that DLOY relates to DI. Afshardoost [71] found that DI positively influenced behavioral intention. Likewise, Kaplanidou and Vogt [43] noted that DI had a significant impact on intention to revisit the destination. According to Kaplanidou et al. [15], DI has a positive impact on behavioral intention and WOM recommendation. Indeed, intention to revisit a destination and positive WOM are principal factors of DLOY [72]. Thus, the following hypothesis was proposed:

**Hypothesis 5 (H5).** *DI will positively influence DLOY.*

SAT is an essential factor in repeat purchase and positive WOM [73,74]. Mechinda [75] found that SAT positively influenced, among tourists, intention to pay more, perceived ESQ, and DLOY. In recent tourism studies, scholars have paid more attention to the association between SAT and DLOY, generally finding that a strong link exists between them [39,40]. SAT influences destination choice, intention to revisit the same destination, and WOM recommendation [16,18]. According to Nilplub [12], if a tourist is content with a destination, he or she is more willing to revisit the place and recommend it to others. Therefore, the following hypothesis was proposed:

**Hypothesis 6 (H6).** *SAT will positively influence DLOY.*

### 2.5. Mediating Effects of ESQ, DI, SAT, and DLOY

With respect to the mediating effect of DI on the relationship between ESQ and DLOY, previous findings indicate that ESQ had a positive impact on DLOY [32,76]. Favorable experiences with service at a destination might lead to repeat visits and positive WOM [6]. Findings from tourism studies suggest that DI influences destination choice. Because previous findings indicate that ESQ is a key driver of DI, which, in turn, affects DLOY, the following hypothesis was proposed:

**Hypothesis 7a (H7a).** *DI will mediate the relationship between ESQ and DLOY.*

Previous findings also suggest that ESQ positively affects SAT and that SAT is a direct antecedent of DLOY [12]. Therefore, the following hypothesis was proposed:

**Hypothesis 7b (H7b).** *SAT will mediate the relationship between ESQ and DLOY.*

Regarding the mediating effect of SAT on the relationship between DI and DLOY, Chi and Qu [10] found that DI indirectly affected DLOY via SAT. Accordingly, the following hypothesis was proposed:

**Hypothesis 7c (H7c).** *DI and SAT will mediate the relationship between ESQ and DLOY.*

Correspondingly, the conceptual research model is shown in the following (Figure 1).

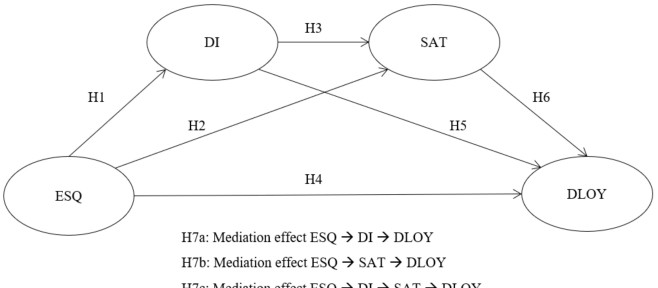

**Figure 1.** Research model. Note. ESQ = event service quality, DI = destination image, SAT = destination satisfaction, DLOY = destination loyalty.

## 3. Methods

### 3.1. Research Site

HengQin (HQ) is an island in South China with an area of only 106.46 square kilometers and about 4000 residents. Due to its natural geographical advantage of being adjacent to Hong Kong and Macao, HQ plays an important role in connecting Hong Kong, Macao, and the mainland. Because this position promotes common prosperity and development, the Chinese government approved the establishment of HQ New Area in 2009, and in March 2019, the State Council of China issued a policy to accelerate, on HQ, the construction of a modern industrial system driven by leisure tourism, including a world leisure tourism center. HQ's tourism has entered a golden period. In 2018, Macao's inbound tourists exceeded 35 million, and the total number of HQ tourists exceeded 15 million.

As an important supply component of tourism destinations, marathon events attract runners and their companions as active sport participants and tourists, respectively. Meanwhile, "smaller recurring sport events, such as local marathons might produce more sustainable economic impact on host communities [25]". In 2018, HQ began using marathons to generate investment and tourism for the city, and in 2019, HQ-M happened for the second year running. As event organizers and destination marketers expected, HQ and HQ-M share a spirit of challenge and continual development. As a national pilot Free Trade Zone, HQ is likely to continue down the path of growth by hosting marathons.

HQ-M is the only professional marathon event in Zhuhai certified by CAA. It welcomes 16,000 runners: 4000 participants for the full marathon, 6000 for the half marathon, and 6000 for the mini-marathon. Regarding event scale, HQ-M is medium to large, relative to marathon events in China. HQ-M has many unique characteristics, including when the event takes place, its cultural focus, its full use of Chinese and Western holidays, and the inclusion of festival activities. To a large extent, HQ differs from other city marathon events in China.

### 3.2. Data Collection Procedure

Data were collected for the current study from repeat participants in the 2019 HQ-M in Zhuhai City, Guangdong Province. Face-to-face interviews and questionnaire-based surveys were conducted on 29 December 2019 with the help of the Organization Committee. The convenience sample in this study included 400 HQ-M participants staying at hotels in the host city or nearby cities for at least one night. Eight questionnaires were eliminated due to incompletion, and the final sample size was 392.

### 3.3. Measures

A seven-point Likert-type scale ranging from strongly disagree (1) to strongly agree (7) was adopted to measure the five perspectives of marathon participants: (a) ESQ, (b) DI, (c) SAT, (d) DLOY, and (e) demographic background. Event service quality was assessed by using 9 items (3 items addressed information quality, 3 interaction quality, and 3 facility quality). These items were adapted from Theodorakis et al.'s study [26] and Huang et al.'s study [25]. Destination image was assessed by using 3 items derived from Chen and Phou [39]. Satisfaction was assessed by using 3 items adapted from Albaity and Melhem's study [54]. Destination loyalty was assessed using 3 items from Loureiro and Miranda [77].

### 3.4. Data Analysis

Data analysis included four stages. First, a descriptive analysis of the questionnaires was conducted to identify the demographic characteristics of the participants and general information about the items in each construct. Second, confirmatory factor analysis (CFA) was performed to test how well the measured variables represented the constructs and to ensure goodness-of-fit for the measurement model. Third, the relationships among ESQ, DI, SAT, and DLOY were examined by using structural equation modeling (SEM) with AMOS 24.0. Finally, mediation effects were tested.

## 4. Results

### 4.1. Descriptive Analysis

Most respondents were male (73.0%, *n* = 286), aged 20 to 29 years old (44.4%, *n* = 174), married (55.9%, *n* = 219), and working as staff in companies (30.6%, *n* = 120). More than half of the respondents (56.4%, *n* = 221) had one to three years of running experience. Participant types were nearly equally split: full marathoner (34.4%, *n* = 135), half marathoner (32.9%, *n* = 129), and runners-for-fun (32.7%, *n* = 128). Length of stay was also nearly equally split: one night (22.7%, *n* = 89), two nights (23.2%, *n* = 91), three nights (27%, *n* = 106), and more than three nights (27%, *n* = 106).

#### Measurement Model

The measurement model was assessed by using maximum likelihood estimation (MLE) in terms of factor loadings, reliability of measurement, convergent validity, and discriminant validity. A preliminary CFA was conducted in the first place. The model fit for CFA was acceptable (ML$\chi2$ = 108.83, df = 59, Normed Chi-square = 1.84, CFI = 0.973, TLI = 0.964, RMSEA = 0.046, SRMS = 0.035). As the standardized factor loading of INT1 did not meet the minimum criterion of 0.5 [76], it was eliminated from INT. Another CFA was then performed. Table 1 shows that all standardized factor loadings of the remaining items ranged from 0.695 to 0.794, indicating that all had convergent validity. The composite reliability of the constructs ranged from 0.753 to 0.820, exceeding the 0.7 value recommended by Nunnally and Bernstein [78] and indicating that the constructs had internal consistency. Finally, all average variance extracted (AVE) values ranged from 0.504 to 0.563, exceeding the recommended 0.5 and indicating that all constructs had adequate convergent validity.

**Table 1.** Results of confirmatory factor analysis.

| Construct | Item | Significance of Estimated Parameters | | | | | Item Reliability | Construct Reliability | Convergence Validity |
|---|---|---|---|---|---|---|---|---|---|
| | | Std. | Unstd. | S.E. | Unstd./S.E. | *p*-Value | SMC | CR | AVE |
| ESQ | ESQINF | 0.730 | 1.000 | | | | 0.533 | 0.753 | 0.504 |
| | ESQINT | 0.695 | 0.965 | 0.078 | 12.372 | 0.000 | 0.483 | | |
| | ESQFAC | 0.704 | 0.943 | 0.083 | 11.333 | 0.000 | 0.496 | | |
| DI | DI1 | 0.762 | 1.000 | | | | 0.581 | 0.820 | 0.532 |
| | DI2 | 0.728 | 0.960 | 0.073 | 13.206 | 0.000 | 0.530 | | |
| | DI3 | 0.719 | 0.872 | 0.065 | 13.444 | 0.000 | 0.517 | | |
| | DI4 | 0.708 | 0.885 | 0.070 | 12.652 | 0.000 | 0.501 | | |
| SAT | SAT1 | 0.696 | 1.000 | | | | 0.484 | 0.766 | 0.522 |
| | SAT2 | 0.736 | 1.186 | 0.102 | 11.632 | 0.000 | 0.542 | | |
| | SAT3 | 0.734 | 1.067 | 0.094 | 11.382 | 0.000 | 0.539 | | |
| DLOY | DL1 | 0.736 | 1.000 | | | | 0.542 | 0.794 | 0.563 |
| | DL2 | 0.718 | 0.975 | 0.079 | 12.389 | 0.000 | 0.516 | | |
| | DL3 | 0.794 | 1.170 | 0.085 | 13.709 | 0.000 | 0.630 | | |

Note. Unstd., unstandardized factor loadings; Std, standardized factor loadings; SMC, square multiple correlations; CR, composite reliability; AVE, average variance extracted.

In addition, Table 2 shows discriminant validity, comparing the square root of the average variance extracted (AVE) with the correlations between the constructs. All of the constructs met the test of discriminant validity because all of the numbers on the diagonal are greater than the off-diagonal numbers.

**Table 2.** Discriminant validity for the measurement model.

| | AVE | ESQ | DI | SAT | DLOY |
|---|---|---|---|---|---|
| ESQ | 0.504 | 0.71 | | | |
| DI | 0.532 | 0.562 | 0.729 | | |
| SAT | 0.522 | 0.625 | 0.566 | 0.722 | |
| DLOY | 0.563 | 0.647 | 0.606 | 0.613 | 0.75 |

### 4.2. Structural Equation Modeling Analysis

In order to examine the hypothesized relationships of the proposed conceptual model, the structural model was tested using MLE. Kline [79] and Schumacker and Lomax [80] recommended nine standard indicators to estimate goodness-of-fit. Table 3 presents several model fit indicators as well as recommended thresholds: $\chi^2$/DF = 1.925 (below the required value of 3), RMSEA = 0.049, SRMR = 0.034 (both less than the required value of 0.08), TLI (NNFI) = 0.962, CFI = 0.971, GFI = 0.942, and AGFI = 0.924 (all above the required value of 0.9). All of the indicators satisfy the independent and combination rules for a good fit.

**Table 3.** Model fit indexes resulted from the SEM analysis.

| Model Fit | Criteria | Model Fit of Research Model |
|---|---|---|
| ML$\chi^2$ | The smaller, the better | 113.563 |
| DF | The larger, the better | 59.000 |
| Normed Chi-square ($\chi^2$/DF) | $1 < \chi^2/DF < 3$ | 1.925 |
| RMSEA | <0.08 | 0.049 |
| SRMR | <0.08 | 0.034 |
| TLI (NNFI) | >0.9 | 0.962 |
| CFI | >0.9 | 0.971 |
| GFI | >0.9 | 0.942 |
| AGFI | >0.9 | 0.924 |

### 4.3. Hypotheses Testing

As shown in Table 4; Table 5, the results support all of the hypotheses featuring direct effects. ESQ is positively related to DI (=0.678, $p < 0.001$), SAT (=0.487, $p < 0.001$), and DLOY (=0.307, $p < 0.001$), supporting H1, H2, and H4, respectively. DI had a positive impact on SAT (=0.286, $p < 0.001$) and DLOY (=0.211, $p < 0.001$), supporting H3 and H5, respectively. SAT positively related to DLOY (=0.205, $p = 0.003$), supporting H6. The squared multiple correlations (SMC = R2) indicate that ESQ explained 31.5% of the DI constructs; both ESQ and DI explained 45.8% of the SAT constructs; and ESQ, DI, and SAT explained 53.7% of the DLOY constructs.

**Table 4.** Path coefficient results in the SEM analysis.

| DV | IV | Unstd | S.E. | Unstd./S.E. | *p*-Value | Std. | R$^2$ |
|---|---|---|---|---|---|---|---|
| DI | ESQ | 0.670 | 0.085 | 7.887 | 0.000 | 0.562 | 0.315 |
| SAT | ESQ | 0.487 | 0.090 | 5.417 | 0.000 | 0.449 | 0.458 |
| | DI | 0.286 | 0.069 | 4.127 | 0.000 | 0.314 | |
| DLOY | ESQ | 0.307 | 0.075 | 4.065 | 0.000 | 0.338 | 0.537 |
| | DI | 0.211 | 0.056 | 3.777 | 0.000 | 0.278 | |
| | SAT | 0.205 | 0.069 | 2.983 | 0.003 | 0.245 | |

Note. DV, dependent variables; IV, independent variables; Unstd., unstandardized regression weights; S.E., standard error; Std, standardized regression weights; R$^2$, coefficient of determination.

**Table 5.** Hypothesis testing findings from the SEM analysis.

| | Constructs of Measurement | Standardized Path Coefficient | Result |
|---|---|---|---|
| H1 | ESQ→DI | 0.678 *** | Supported |
| H2 | ESQ→SAT | 0.487 *** | Supported |
| H3 | DI→SAT | 0.286 *** | Supported |
| H4 | ESQ→DLOY | 0.307 *** | Supported |
| H5 | DI→DLOY | 0.211 *** | Supported |
| H6 | SAT→DLOY | 0.205 ** | Supported |

Note ** $p < 0.01$, *** $p < 0.001$.

### 4.4. Mediation Effects

Bootstrapping was used to perform mediation analysis and examine indirect effects. The results for indirect effects in Table 6 support all of the hypothesized mediation relationships. In summary, all of the nine hypothesized paths were evident in the conceptual structure of ESQ, DI, SAT, and DLOY, including six direct paths (see Table 5) and three indirect paths (see Table 6). ESQ had both direct and indirect effects on SAT and DLOY. DI had both direct and indirect effects on DLOY. These paths indicate that the constructs of ESQ, DI, and SAT are important antecedents of DLOY. In particular, ESQ (β = 0.588, $p < 0.001$) had a greater effect on DLOY than DI (β = 0.269, $p < 0.001$) and SAT (β = 0.205, $p < 0.01$). However, DI and SAT mediated the relationship between ESQ and DLOY and showed significant serial mediation as well. The effect of DI was greater than the effect of SAT (see Table 5; Table 6).

**Table 6.** Mediation effect analysis from the SEM analysis.

| | Hypothesis | Standardized Path Coefficient | Standard Error | 95% Asymmetric Confidence Interval | Result |
|---|---|---|---|---|---|
| H7a | ESQ→DI→DLOY | 0.142 | 0.046 | (0.065, 0.241) | Supported |
| H7b | ESQ→SAT→DLOY | 0.100 | 0.042 | (0.065, 0.241) | Supported |
| H7c | ESQ→DI→SAT→DLOY | 0.039 | 0.020 | (0.011, 0.089) | Supported |

### 5. Discussion and Conclusions

The aim of the current study, based on previous discussions and investigations of the inter-relationships among ESQ, DI, SAT, and DLOY, explored the causes of DLOY among repeat participants in a city marathon event. The findings extend a common understanding of the relationship between sport event marketing and city marketing in four ways. First, repeat runners and first-time participants were distinguished and focused on the former in the 2019 HQ-M, for active sport tourists are to some extent different from passive sports tourists in attitude and behavior [15]. Using an exclusive sample of repeat runners, how the constructs of ESQ, DI, and SAT might predict DLOY behavior was investigated. While some scholars, such as Jeong et al. [14], have collected data from spectators and runners at city marathons to examine the relationship between event quality and DLOY, none have examined the determinants of DLOY among repeat participants in the context of sport event marketing and city marketing. Second, instead of measuring the separate effects of ESQ, DI, or SAT on DLOY and ignoring the mediating effects of DI and SAT [10,81], an integrated structural model was tested including all four constructs, producing new findings for sport event marketing and city brand sustainability. Third, ESQ was found to play an essential role in achieving DLOY among marathon runners. Based on the total effect coefficient, ESQ appeared to have the most important effect on DLOY. Additionally, four paths emerged from ESQ to DLOY: direct, mediated by DI, mediated by SAT, and mediated serially by DI and SAT. These results suggest ESQ is crucial to take into account when assessing event participant DLOY, as previous scholars have found [14,15,56]. Fourth, the findings highlight the mediation effect and dual mediation effect (i.e., serial mediation of DI and SAT between ESQ and DLOY). In particular, DI had a greater mediation effect than SAT. According to the push–pull theory, DI is the external push force, and SAT the internal pull force, which leads to the choice of destination and DLOY. This idea implies that DI and SAT are worthy of more attention, even though ESQ plays a key role in predicting DLOY. Active sport tourists (e.g., marathon runners) primarily differ from general tourists because the DLOY of the former relates to the sporting event [9]. Timely and sufficient information and service from the organizer will affect registration and event participation, revisit intention, WOM recommendation, and willingness to encourage friends and relatives to visit the host city.

As city marathons face fierce competition and greater challenges, knowing why runners participate in city marathons and what drives their loyalty to the host city is essential. Findings in this study confirm that high ESQ, favorable DI, and high SAT

enhanced DLOY separately and simultaneously. The findings have important implications for strategic destination management and can help city managers design and implement marketing programs to improve ESQ, DI, and SAT. First, ESQ is a key factor for event and destination managers to consider because it is a direct antecedent of DI and SAT and influences DLOY. Event marketers and destination managers must strive to improve ESQ if they want to compete successfully in the city marathon market, strengthen revisit intention, and increase WOM recommendation. Given the three dimensions of ESQ, city marketers and event organizers need to understand what event participants need in terms of INF, INT, and FAC and then prioritize those needs when organizing events. When planning marathon events, organizers should pay attention to information service quality. Runners often want to search for event and destination information using mobile apps, check the record after running and training, and send photos via social media. Marketers need to understand these information habits and provide high-quality and timely event publicity on the internet and social media. Regarding facilities at a marathon event, repeat participants most desire shuttle buses, parking lots, water and aid stations, and convenient facility layout. Improving ESQ can enhance DI, SAT, and, eventually, DLOY. Although simultaneously controlling all of the elements that contribute to ESQ is difficult, managing some of them (e.g., advertising, event information offices, routine planning, shuttle buses, volunteers, websites, public relations, and tour agents) is feasible. Furthermore, DI is an important factor in marketing a sporting event and the host city. DI might have an even greater positive effect on DLOY than SAT. Therefore, city marketers and event organizers should improve the four aspects of DI: atmosphere, natural environment, entertainment, and exoticness of the host city. They should also develop the unique attributes of an event and the destination to attract participants and build a sound brand. Doing so is crucial for the host city to gain a competitive advantage in an increasingly challenging market. Moreover, event and destination marketers should pay more attention to SAT. SAT was found to be positively related to DLOY, implying that higher SAT might lead to higher loyalty to the host city. High ESQ is likely to increase SAT, and poor ESQ likely to leave tourists dissatisfied. In order to retain marathon runners, managers and marketers need not only to improve ESQ but also to provide local food, public transportation, souvenirs, and accommodations in order to strengthen the host city's image and loyalty.

The current study also has some limitations. First, the marathon industry is soaring high in China; however, most of the marathon events have a short history. HQ-M has happened only twice. The relationships among ESQ, DI, SAT, and DLOY might depend on the amount of time a marathon has been around. In the future, scholars should compare marathons with different length histories, as well as different sporting events, to confirm the interrelationships among these variables. Second, scholars should explore the predictive validity of the scale used to assess ESQ of recurring sporting events, DI, SAT, and loyalty to the host city.

**Author Contributions:** Conceptualization, X.C., L.Z. and J.J.Z.; Data curation, Z.T. and T.L.; Formal analysis, X.C., B.H.Y.; Funding acquisition, X.C.; Investigation, X.C., Z.T. and T.L.; Methodology, X.C., B.H.Y. and J.J.Z.; Project administration, X.C., L.Z.; Resources, X.C., T.L.; Supervision, L.Z., J.J.Z.; Writing—original draft, X.C., Z.T. and L.Z.; Writing—review & editing, B.H.Y., J.J.Z. and Z.T. All authors have read and agreed to the published version of the manuscript.

**Funding:** This research received the funding of China's National Social Science Research Funding (17BTY055).

**Informed Consent Statement:** Informed consent was obtained from all subjects involved in the study.

**Data Availability Statement:** This questionnaire-based survey and data are true, objective and valid. We conducted this survey with the assistance of the Organization Committee of Hengqin Marathon (HQ-M) in Zhuhai City, Guangdong Province.

**Conflicts of Interest:** The authors declare no conflict of interest.

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
