# Peer review of "“One Event, One City”: Promoting the Loyalty of Marathon Runners to a Host City by Improving Event Service Quality"

_sustainability, doi:10.3390/su13073795_

Round 1

Reviewer 1 Report

The article has only minor ammendemnts necessary, namely Table 1 way of presentation that is not clear.The 4 elements tested ESQ, DI, DS, and DLOY, seem well and results comprouved therir importance.

Author Response

“One Event, One City”: Promoting the Loyalty of Marathon Runners to a Host City by Improving Event Service Quality

Revision on March 15, 2021

Response to the Reviewer’s Comments

The article has only minor amendments necessary, namely Table 1 way of presentation that is not clear. The 4 elements tested ESQ, DI, DS, and DLOY, seem well and results comprised their importance.

Response: Thank you for your positive review and support of our manuscript. In this revision, we have made strong efforts to reduce the confusion issue pointed by you by removing Table 1 and incorporating the information into the text (please see page 6, line 255-263 in the original manuscript). We hope that this alternative solution is acceptable to you.

Please see section 3.3. Measures: A seven-point Likert-type scale ranging from strongly disagree (1) to strongly agree (7) was adopted to measure the five perspectives of marathon participants: (a) ESQ, (b) DI, (c) SAT, (d) DLOY, and (e) demographic background. Event service quality was assessed by using 9 items (3 items addressed information quality, 3 interaction quality, and 3 facility quality). These items were adapted from Theodorakis et al.’s study [26] and Huang et al.’s study [25]. Destination image was assessed by using 3 items derived from Chen and Phou [39]. Satisfaction was assessed by using 3 items adapted from Albaity and Melhem’s study [55]. Destination loyalty was assessed using 3 items from Loureiro and Miranda [78].

 Please also see the attachment. Thank you!

Reviewer 2 Report

Authors present a very interesting research about the importance of hosting sport events like marathons to promote city management, tourism, economic activities, planning and development. They define various parameter to study this new phenomenon in China: destination loyalty, destination image, satisfaction and event service quality. They apply the method to Heng Qin island and its marathon in 2019. The presented methodology, as described in text and in Figure 1, is clear and, in my opinion, can be used in different context. One of the most important aspects is the use of qualitative, perceptive and subjective data to analyze in an objective way the issues. The paper is well structure in all the sections, but I do not agree with the "WE" form for a scientific journal as Sustainability is. Please revise.

Moreover, I suggest to insert more references for the first part of the introduction section (from line 37 to line 47) and to put chapters 6 and 7 in Discussion and Conclusion part to make a more homogeneous and complete text.

Author Response

“One Event, One City”: Promoting the Loyalty of Marathon Runners to a Host City by Improving Event Service Quality

Revision on March 15, 2021

Response to the Reviewer’s Comments

Authors present a very interesting research about the importance of hosting sport events like marathons to promote city management, tourism, economic activities, planning and development. They define various parameters to study this new phenomenon in China: destination loyalty, destination image, satisfaction and event service quality. They apply the method to Heng Qin Island and its marathon in 2019. The presented methodology, as described in text and in Figure 1, is clear and, in my opinion, can be used in different context. One of the most important aspects is the use of qualitative, perceptive and subjective data to analyze in an objective way the issues. The paper is well structure in all the sections, but I do not agree with the "WE" form for a scientific journal as Sustainability is. Please revise.

Response: Thank you for pointing out positive aspect of our manuscript. In this revision, we have made strong efforts to address your concerns and suggestions. We have switched “We” into the “passive” form throughout the paper. We hope that our revision is satisfactory.

Moreover, I suggest to insert more references for the first part of the introduction section (from line 37 to line 47) and to put chapters 6 and 7 in Discussion and Conclusion part to make a more homogeneous and complete text.

Response: Thank you for pointing this issue out. Yes, we agree to add more references to the first part of the introduction. On page 1 (Please see Section 1 Introduction), we have added the following:

After the 2008 Beijing Olympic Games, China entered a new stage by that the government began promoting national fitness [1], the sports industry [2], and city tourism [3], simultaneously. Since 2014, the Chinese government has successively enacted more than 18 policies to include the sports industry as a part of the efforts of transforming and upgrading the national economy. In 2016, the State Council of China released an outline for “Healthy China 2030,” an initiative to promote health among all citizens [4].

1.    Wang, X. L. Research on the Development Trend of Mass Sports in China after Beijing Olympic Games, Sports Culture Guide, 2008,11, 4-5.

2.    Yu, Q. and Yuan, J. New Mentality of Post—Olympic Games Period on Country Sports Industrial Development. Journal of Sports and Science.2009,30(4), 7-10.

3.    Chen, F. Research on Strategy for Sustainable Development of Chinese Sports Tourism. Journal of Sports and Science.2010,31(5), 44-49.

4.    The State Council of the People’s Republic of China. The Central Committee of the Communist Party of China and the State Council issued the "Outline of the "Healthy China 2030" Plan”. http://www.gov.cn/xinwen/2016-10/25/content_5124174.htm

The chapter 6 and 7 were merged into the Discussion and Conclusion chapter as you suggested. Thank you!

 Please also see the attachment. Thank you!

This manuscript is a resubmission of an earlier submission. The following is a list of the peer review reports and author responses from that submission.